# Thermal Behavior of Tropical Sea Cucumber of *Isostichopus isabellae*: Preliminary Issues

**DOI:** 10.3390/ani14243613

**Published:** 2024-12-15

**Authors:** Adriana Rodríguez-Forero, Jose Villacob-Royerth, Mónica Hernández Rodríguez

**Affiliations:** 1Grupo de Investigación y Desarrollo Tecnológico en Acuicultura—GIDTA—Fisheries Engineering Program, Universidad del Magdalena, Santa Marta 470004, Colombia; 2Departamento de Acuicultura, Centro de Investigación Científica y de Educación Superior de Ensenada (CICESE), Ensenada 22860, B.C., Mexico; mhernand@cicese.mx

**Keywords:** *Isostichopus*, sea cucumber, growth, thermal behavior, thermal resistance, tropical

## Abstract

We assessed how the sea cucumber *Isostichopus isabellae* grows and handles different temperatures. We kept 60 sea cucumbers in 250 L tanks at 23 °C and 26 °C for 30 days to see which temperature was best for them. We looked at how well they survived and grew. To see which temperature they liked best, we placed them in a setup with a thermal gradient of > 8 ranging from 20 °C to 29 °C and observed their behavior for 4 h. They chose the temperature they were used to, matching the best temperature found in our study. There were no great differences in weight loss between the two test temperatures. The highest survival rate, 86%, was observed at 23 °C, making it the best temperature for maintaining the sea cucumbers. We found the highest and lowest temperatures they could handle by slowly heating or cooling the water until they showed signs of stress. The highest temperature they could take was 36.5 ± 0.3 °C, while the lowest was 8 ± 0.5 °C. Based on these findings, *Isostichopus isabellae* could be negatively affected by higher temperatures, so we recommend keeping them at a temperature of 23 ± 2.3 °C in captivity for tropical aquaculture.

## 1. Introduction

Water temperature is one of the most important factors that affects growth, metabolic rate, and other physiological processes in aquatic ectotherm animals [1]. Wood and McDonald [2] and Nithirojpakdee and co-workers [3] highlighted that aquatic organisms have physiological, behavioral, and genetic characteristics that allow them to confront environmental conditions, called acclimatization, a component of short-term reversible physiological adaptation. This depends on the individual’s previous experience [4,5].

Studies of thermoregulatory behavior, specifically the optimum temperature, describe it as the temperature at which an organism optimally performs some biological processes, such as growth, reproduction, metabolism, swimming speed, and maximum cardiac work, among others [6,7,8]. Therefore, the temperature selected by organisms in a thermal gradient is known as the final preferendum, which was defined by Fry (1947) [9] as the “temperature around which organisms congregate when placed in a thermal gradient regardless of their previous thermal history and that preferred temperature that is equal to the acclimation temperature”. This thermal response can be modified by factors such as age, sex, food availability, seasonality, pathological conditions, and inter- or intra-specific competition, among others [10,11].

Other responses evaluated in aquatic animals include their tolerance and resistance to temperature. Incipient upper and lower lethal temperatures characterize the thermal tolerance of aquatic organisms [12,13], and the critical thermal minimum (CTMin) and maximum (CTMax) define the resistance zone [12,14,15]. The upper incipient lethal temperature (UILT) and lower incipient lethal temperature (LILT) define the tolerance zone of a species. The limits of the ULT and LLT are represented by the mean lethal temperature (TL50), where 50% of a population theoretically can survive indefinitely [16]. Critical temperatures can be recorded experimentally in a very short time (less than two hours) and reveal biotic or abiotic factors’ influences on this response [17]. As critical temperatures, maximum (CTMax) or minimum (CTMin), are related to the natural environment, they have been used as indicators of stress and adaptation [18], characterizing the thermal resistance of an organism by identifying a specific response which occurs at a point of temperature [19], in addition to reflecting its capacity to survive in environments with extreme temperatures [20].

In the phylum Echinodermata, research has been carried out with this approach by few authors. Hernández co-workers [21] determined the critical thermal maximum and its effect on the osmotic pressure of the body fluid of the red sea urchin *Strongylocentrotus franciscanus*. An and co-workers [22] evaluated the effects of temperature and food ration on growth in juveniles of the sea cucumber *Apostichopus japonicus*, while [23] studied the thermal resistance and expression of Hsp70 in the same species, finding higher lethal temperatures between 30 and 31.8 °C. The effect of temperature on the performance of some cucumber species has also been evaluated. In *A. japonicus*, the effects of different thermal regimes on growth and physiological performance were evaluated [24]. The authors highlighted that small and medium temperature fluctuations accelerated the growth of sea cucumbers, and large temperature fluctuations diminished this growth. Wang co-workers [25] evaluated the effects of different rearing temperatures on the growth, metabolic performance, and thermal tolerance of juveniles of the sea cucumber *A. japonicus*. The authors found that *hsps* expressions increased with a rise in rearing temperatures, which, in turn, would explain the decrease in the growth rate of sea cucumbers at a high temperature. The CTmax was positively correlated with the rearing temperature. Sun and co-workers [26] evaluated the role of water temperature in controlling the feeding and locomotion behavior and digestive physiology in the sea cucumber *A. japonicus*, highlighting the importance of determining its preferred water temperature conditions to assist in the design of suitable holding conditions for the captive breeding of this species.

Sea cucumber belong to the phylum Echinodermata and constitute one of the main groups that is marketed and consumed in the Asian countries, with China being the largest consumer of this resource [27]. In addition to their commercial value, sea cucumbers have great ecological importance as modifiers of substrates, ingesting sediments, modifying their composition, and recycling organic matter as a biofilter [28,29,30]. Because of their commercial value due to their nutritional and medicinal properties and ecological role [31,32], sea cucumbers are an important species, and knowledge of their basic biology must be deepened. In this regard, information on tropical marine cucumber species native to the Colombian Caribbean Sea is scarce. Although eating cucumbers is an uncommon practice in the study area, there are numerous species here, some of which, like *Isostichopus isabellae*, may be potential candidates for sustainable production and addressing the challenges associated with their commercialization in international markets. In this species, different studies have been conducted, leading to their production in captivity, including topics such as its gametogenesis, reproduction in captivity, proximal composition, densities, and diets, among others, which gradually consolidate their culture technology [33,34,35,36,37,38,39,40,41]. Considering that, during the second half of the year, the temperatures of the Caribbean Sea rise, this period is particularly suitable for implementing induced breeding programs in captivity [34,40,41].

The objective of this study was to determine the most favorable temperature for conditioning based on growth, survival, temperature preferences, and stress responses to increasing and decreasing temperatures to characterize the thermal tolerance of the sea cucumber *Isostichopus isabellae*. To meet this goal, we described the behavior of the cucumbers when exposed to different conditions of temperature.

## 2. Materials and Methods

### 2.1. Ethics Statement

Sea cucumber handling was carried out in accordance with the Universidad del Magdalena guide for the care and use of laboratory animals. The animal study was reviewed and approved by the Ethical Committee of the Universidad del Magdalena, which assesses animal care in research activities, and according to the ethical standards of the Guide for the Care and Use of Laboratory Animals [42].

### 2.2. Biological Samples

*Isostichopus isabellae* [39] were collected in Rodadero’s Bay (11°12′32″ N, 74°14′12″ W), an area where extreme temperatures range from 20 to 29 °C [43,44,45]. This interval was considered for developing the assays, which are described below. One hundred and sixty animals were transported in tanks (20 L) filled with seawater at 20 °C, cooled with frozen water bottles.

Once in the laboratory facilities of Aquaculture at the Universidad del Magdalena, the sea cucumbers were placed in 250 L tanks with 12 animals each and a temperature of 26 °C, where they were fed with a mixture of sediment (consisting of white-gray sand composed of coarse to medium-fine sand (no more than 10 mm thick) and *Tetraselmis* sp. (5.5 × 10^6^ cell mL^−1^ day^−1^).

The specimens came from species that were recorded previously in the Center of Biological Collections of the Universidad del Magdalena (Santa Marta, Magdalena, Colombia) (CBUMAG:ECH:00001, CBUMAG:ECH:00002, and CBUMAG:ECH:00003). Their morphology was examined and compared, and their genetic structures were acknowledged through 16S rDNA and COI data [39].

### 2.3. Sea Cucumber Favorable Temperature of Conditioning of the Culture

The sea cucumbers were acclimated to two temperatures (23 and 26 °C) for 30 days in 250 L tanks with filtered seawater (salinity 36 psu) by 1µ, sterilized with UV, and maintained with constant aeration. The cucumbers were weighed on a scale (Brand Ohauss of 0.1 g) and distributed randomly. Ten sea cucumbers were stocked per tank, and during acclimation, the organisms were fed with the same mixtures of sediment and *Tetraselmis* sp. (5.5 × 10^6^ cell mL^−1^ day^−1^) provided daily by the microalgae production laboratory of the GIDTA research group cultured in Guillard F/2 medium [46]. Daily maintenance was performed, and a 40% water exchange every two days was performed to preserve the water quality; mortality was recorded daily.

The favorable maintenance temperature was determined based on the growth and survival of the sea cucumbers during cultivation. Sixty adults (196.81 ± 63.05 g, mean ± SD) were exposed to 23 °C and 26 °C. Thirty animals in each selected temperature (ten specimens per tank) were located in three tanks (250 L), with three replicates per treatment. Samplings were carried out every 10 days, and the animals were weighed to evaluate changes in their weight until the end of the experiment. The weight gain (%), specific growth rate (SGR), and coefficient of variation for body weight (C.V.) were calculated using the equations proposed by [47,48].
SGR (% d − 1) = 100 (lnW2 − lnW1)/D,
CV (%) = 100 × (SD/X)
where W1 and W2 are the initial and final body weights of the sea cucumbers (g); D is the duration of the experiment; SD is the standard deviation in body weight; and X is the mean weight (g) for a particular sampling period.

Survival (%) was tested every day of the experimental period and is expressed as the percentage of the initial number of cucumbers stocked.

During this test, the cucumbers were fed with a mixture of sediment and *Tetraselmis* sp. (5.5 × 10^6^ cel mL^−1^ day^−1^). Daily maintenance was performed by partial water exchange (30% water volume) to preserve the water quality. The conditioning temperatures were maintained using an Inverter air conditioner (12,000 BTU) for the 23 °C condition and heaters (Sunlike, 100 W, Xilong, China.) with an electronic thermostat for the 26 °C condition. Temperature was measured twice a day with a HACH LANGE 5130 thermometer and water pH with a HACH LANGE 5130 pH meter, while oxygen concentration was measured with a HACH 85004 and salinity with a HACH 5500 (Table 1).

### 2.4. Sea Cucumber Thermal Preference

#### Rearing System and Experimental Conditions

The sea temperature in Santa Marta Bay has a range from around 20 to 29 °C [43,44,45], so we considered this range for the study of the thermal preferendum of the sea cucumbers. To assess the thermal preferendum of the sea cucumbers, a temperature channel was built based on the modified design by [49]. For this purpose, we used 3 m of horizontal PVC piping (6-inch diameter), and two Sunlike 100 W quartz heaters with an electronic thermostat were placed at one end and bags of frozen hydrogel were placed at the opposite end to achieve the thermal gradient. Ten digital thermometers with a thermocouple (TPM-10) were located along the chamber to monitor the temperature gradient. An air diffuser hose placed along the channel created enough circulation to prevent thermal stratification of the water. The system was tested over three days before the experiment, recording the temperature every 15 min for 3 h to ensure that the temperatures within the gradient could be maintained within the limits from 20° to 29 °C.

Twenty-four cucumbers (177.40 ± 75.21 g) were divided into two groups of twelve individuals, and then acclimated over 20 days at 23° and 26 °C in 250 L tanks. Three sea cucumbers were exposed and placed in the chamber of the thermal gradient, matching the same acclimation temperature. The trials began 30 min later to reduce handling stress. The temperature and sea cucumber location in the gradient cameras were registered every 15 min over four hours in a horizontal thermal gradient (Figure 1). All experiments were performed in triplicate, using three animals each time, separately for each acclimation temperature.

### 2.5. Sea Cucumber Thermal Resistance

Critical temperatures are characterized by a response indicating that the activity of organisms begins to be disorganized, and they lose the ability to escape from conditions that will quickly lead to death [12,14]. In this study, due to a lack of organisms to evaluate the maximum and minimum critical temperatures in both acclimation temperatures, we only characterized the sea cucumber responses that described the TCMin at the temperature of 23 °C and the TCMax at the temperature 26 °C when decreasing or increasing the water temperature, respectively.

Two groups with twelve sea cucumbers each were acclimated at 23 and 26 °C. These temperatures were established based on data obtained in the study area to determine the critical maximum (CTMax) and minimum (CTMin) temperatures of the animals. These trials were run in triplicate for each acclimation temperature. Twenty-four hours before the experiment, the animals were not fed to reduce their metabolic waste during the trial.

#### 2.5.1. Critical Thermal Maximum (CTMax)

Four sea cucumbers (230 ± 10 g, mean ± SD) were placed in 18 L aquariums with two quartz heaters (Sunlike 25 W) attached to an aeration tube to distribute the heat. The water temperature at 26 °C was increased at a rate of 1 °C/0.5 h in each trial. The experiment was stopped when 50 percent of the sea cucumbers showed a muscle relaxation response due to exposure to an increased temperature.

#### 2.5.2. Critical Thermal Minimum (CTMin)

Four sea cucumbers (236 ± 13 g, mean ± SD) were placed in aquariums of 18 L with frozen hydrogel bags attached to an aeration tube to distribute the cold. The temperature at 23 °C was decreased at a rate of 1 °C/0.5 h in each trial. The experiment was stopped when 50 percent of the sea cucumbers showed the cessation of body movement due to exposure to a decreased temperature.

### 2.6. Statistical Analysis

Data analysis was performed using the statistical software Statgraphics Centurion XVI version 16.1.18, and data are expressed as mean ± SD. For the sea cucumbers’ favorable maintenance temperature in culture, homoscedasticity was tested with a Levene’s test. The Student’s *t*-test was applied to compare the preferred temperatures of the organisms acclimated to 23° and 26 °C. A two-way ANOVA test was performed to compare the weight gain over time between the two treatments. A one-way ANOVA was performed to compare the temperatures where different responses were observed when increasing or decreasing the temperature considering the acclimation temperatures. All tests were performed with a level of significance of 0.05.

## 3. Results

### 3.1. Favorable Conditioning Temperature of Sea Cucumber

Growth parameters are shown in Table 2. There was a decrease in the weight of the animals at both experimental temperatures (Figure 2). At the end of the test, there was no significant difference (two-way ANOVA, *p* = 0.7) in the final weight between the animals acclimated to 23 and 26 °C. The coefficients of variation for the initial weight of the animals acclimated at 23° and 26 °C were 29.6% and 31.9%, respectively, while those of the final weight were 7.0% and 19.7%, respectively. The survival of the organisms acclimated at 26 °C was 10%, while that in those acclimated to 23 °C was 87.7% (Table 2).

In the period between day 10 and day 20, the weight loss of the organisms was 3.06% in those acclimated to 23 °C and 0.92% in the sea cucumbers acclimated to 26 °C (Figure 2). The weight loss in the first 10 days showed statistically significant differences (*p* < 0.05) between the two periods of 0–10 and 10–20 days at both tested temperatures. There were no statistically significant differences (*p* > 0.05) between the weight loss of the sea cucumbers with respect to temperature from day 10 to the end of the experiment (Figure 3). The negative values of the weight gain and specific growth rate show the net cumulative percentage and percentage per day of weight loss of the sea cucumbers at different temperatures (Table 2).

### 3.2. Thermal Preference of Isostichopus isabellae

Regarding thermal preferences, there were significant differences (*p* = 0.001) relative to the acclimation temperatures (23° and 26 °C) (Figure 4). For the animals acclimated to 23 °C, their preferred temperature corresponded to the higher frequency of temperatures visited in the gradient, which stood at 23 ± 0.23 °C. Sea cucumbers that were acclimated to 26 °C frequented the temperature of 26 ± 0.3 °C for a longer time. During the development of the experiment, the cucumbers showed hardly any activity, so the selected temperature was similar to that of the acclimation temperature.

### 3.3. Thermal Resistance of Isostichopus isabellae

#### 3.3.1. Behavioral Responses to Characterize the Critical Thermal Maximum (CTMax)

The responses observed in the sea cucumbers with an increasing water temperature are described (Figure 5 and Figure 6).

(a)At the beginning of the experiment (time 0), the individuals showed normal behavior representing slow movement. (Figure 5A).(b)The cucumbers were gregarious. No papillae were observed in expansion (Figure 5B).(c)One hour after at 28 ± 0.3 °C, the animals showed larger movements of the podia (8 oscillations/10 sec), with their papillae projected and extended. Also, 75% changed their position to be placed in a group along the wall of the aquarium (Figure 5C).(d)At 30 ± 0.5 °C, the cucumbers were grouped in the corner of the aquarium and stuck to the walls (Figure 5D). This behavior was defined as an increase in movement of podia. Papillae were projected and extended (IMP + PE).(e)At 31 ± 0.3 °C, the cucumbers were dispersed at the bottom of the aquarium, with a decrease in movement of the podia (5 oscillations/10 s), while the length of the animals increased by approximately 1.5 times their initial size (Figure 5E). This response is reflected in the decrease in movement of the podia and the start of corporal relaxation (DMP + SCR).

Relaxation and body contraction (RBC) and body relaxation end with tentacles extended (BRF + TE) were also observed in 100% of the sea cucumbers before the end of the trial (Figure 6). The different behaviors observed in the sea cucumbers acclimated to 26 °C and exposed to an increase in temperature showed significant differences (*p* < 0.05), except between the IMP + PE and DMP + SCR responses.

#### 3.3.2. Behavioral Responses to Characterize the Critical Thermal Minimum (CTMin)

There was a statistically significant difference (*p* < 0.05) between the temperatures, where different responses were observed in the sea cucumbers acclimated to 23 °C. The following describes their responses (Figure 7 and Figure 8), which were registered in sequence as the temperature of the water decreased.

(a)In the beginning (time 0), the animals showed normal behavior (Figure 7A).(b)When the temperature decreased to 20 ± 0.2 °C, the cucumbers began to move toward the walls or toward the corner of the aquarium when possible (Figure 7B). The behavior was defined as the start of the activity (SA).(c)At 18 ± 0.1 °C, the animals overlapped with each other. In total, 60% extended their tentacles and 40% took the position of a “cobra” (Figure 7C). These behaviors are summarized as tentacles extended and cobra body position (TE + CBP).(d)At 14 ± 0.5 °C, the animals took the position of a U shape, were separated from each other, and their movement was virtually nil (Figure 7D). This response was characterized as a total decrease in movement and body position in U (TDM + BPU).(e)When the temperature of the water reached 9 ± 0.2 °C, 62.5% of the animals simultaneously lost the U shape, had slight movements, extended their tentacles, and were not relaxed (Figure 7E). At 8 ± 0.5 °C, there was no movement. These behaviors were summarized as decreased movement of podia, tentacles relaxed, and the cessation of body movement (DMP + TE + CBM).

## 4. Discussion

### 4.1. Favorable Temperature of Maintenance

Studies on the responses of preference, thermal tolerance, and resistance have been widely documented in different vertebrate and invertebrate species, due to their physiological influences, such as on growth and survival, among others [4,6,21,24,50,51,52,53,54,55,56,57]. In sea cucumber species, there have been few studies carried out on their behavioral responses to temperature [23,24,25], so in this research, the favorable maintenance temperature, preferences, and thermal resistance of *Isostichopus isabellae* were documented as a contribution to knowledge on its thermal biology.

*Isostichopus isabellae* saw decreases in its growth, SGR, and FCE indexes at both experimental temperatures. This could have been due to the diet supplied probably not meeting the nutritional requirements of the species during this stage of the culture. The food supplied consisted of marine sludge and microalgae, similar to others supplied to other sea cucumber species [58,59,60,61]. However, compound diets that lead to better growth indicators need to be tested, with the diets supplied being an obstacle to overcome. Currently, large-scale breeding technology for *Isostichopus* in Latin America is not well-established, which limits the development of large-scale artificial farming. One of the key aspects to be explored in depth is dietary requirements and tailored nutrition strategies suited to sea cucumber needs.

In juveniles of *Apostichopus japonicus*, the effects of temperature (16, 18, 20, and 22 °C) and food ration (0%, 0.3%, 0.6%, and 1.4% of body weight per day) on growth (mean body weight 5.4 ± 0.1 g) were evaluated for 35 days [22]. The authors found that the specific growth rate (SGR) and condition factor decreased significantly with an increase in temperature from 16 °C to 22 °C, suggesting that a decrease in SGR can be attributed to food conversion efficiency (FCE), which was lower at high temperatures. In addition, the decrease in growth in juveniles exposed to the highest temperature was attributed to the decrease in the FCE and the increase in energy consumed by breathing, so *A. japonicus* cannot obtain the energy needed when the temperature is above 22 °C.

Although adult-stage sea cucumbers can increase their growth slightly, there was no evidence found in this study. There was an observed weight loss of 39.18% and 27.89% in the sea cucumbers maintained at 23 and 26 °C, respectively. The growth rate did not show positive behavior, so survival was considered to establish which of the two thermal conditions was the most favorable for maintaining sea cucumbers under laboratory conditions. This assertion is based on that fact there were no differences between the weight loss of the cucumbers acclimated to different temperatures from day 10. However, these results are not intended for use in culture, since only two temperatures were evaluated, and we believe that these aspects should be studied when taking into account a wider thermal spectrum.

According to Yanagisawa [62] and Zamora and Jeffs [63], the optimum water temperature for sea cucumbers is specific to the species, which has been determined in a very few holothuroids. In this study, we observed that, although there was no difference in the weight loss of the cucumbers acclimated to 23 and 26 °C, this result might have been masked, because at the end of the assessment period at 26 °C, only three organisms had survived. Maybe heavier cucumbers were able to survive this temperature. These results agree with those reported by Zamora and Jeffs [63], who claimed that an increase in water temperature for the Australasian Sea cucumber *Australostichopus mollis* caused a significant reduction in their food consumption rate and, therefore, a decrease in growth due to the increase in their metabolic rate. These changes could be associated, in turn, with respiratory problems, ion exchange, and osmotic pressure [22,24,53,64]. Ji and co-workers [64] and James [31] stated that sea cucumbers exposed to high temperatures change their eating habits or stop feeding, a behavior that leads to body weight loss and subsequently death, as happened over the course of the present study. For these reasons, our results suggest that 23 °C would be the best water temperature to keep the sea cucumber *Isostichopus isabellae* under laboratory conditions. These results are different to those found for *Apostichopus japonicus* (16 °C) [64] and *Holothuria scabra* (30 °C) [65], thus confirming that the thermal response is species-specific, as previously postulated by Ge and co-workers [61]. In order to formulate a better approach, tropical sea cucumbers such as the species used in our study should be kept in the laboratory for longer, and it is also necessary to continue generating knowledge on the thermal requirements of these species. In their natural habitat, the tropical cucumbers assessed in this study thrive at temperatures ranging from 20 to 26 °C, typically at depths between 5 and 10 m. When acclimatized under controlled conditions for aquaculture practices, they appear to experience the most favorable well-being at approximately 23 °C. This temperature may represent an ideal starting point for larger-scale cultivation efforts. A broader temperature range would provide a more practical conclusion, enhancing the overall development of the species.

### 4.2. Thermal Preference of Isostichopus isabellae

Cossins and Bowler [66] stated that temperature is one of the key items in the mosaic of physical and biotic factors that describe the niche of an organism. In this regard, thermal preference is a response that may vary due to environmental factors other than temperature, which depends on the genetics of the species, so it is considered to be a species-specific response [67]. In turn, this response is determined by the metabolic efficiency of an organism, which affects its best growth, reproduction, and behavior [68,69]. In this study, the thermal preference of *Isostichopus isabellae* maintained at 23 and 26 °C was similar to the temperature at which the animals were acclimated. Several authors have claimed that, for many species, thermal preference is primarily a function of the thermal history of the individuals’ acclimatization or acclimation [6,57,70]. In this context, and taking into account the information reported for various species of aquatic organisms, the tendency observed is that preferred temperatures vary markedly between species in a direct relationship or inversely proportional to the temperature of acclimation [6,71,72]. In the case of sea cucumber, this was not fulfilled, because the sea cucumbers remained at the same acclimation temperature. Perhaps due to the behavior of sea cucumbers, this type of study could be carried out with an observation time greater than 24 h to observe their responses in a thermal gradient.

In *Isostichopus isabellae*, it was not possible to determine the value of their acute thermal preference, because the two trial temperatures intercepted the isothermal line and, during the experiment, the individuals did not move in the established evaluation time. It is important to highlight that the determination of temperature preference could be used as a rapid method for the estimation of the temperature required to promote physiological processes such as the growth of animals [6,7,8] and contribute to their rapid development in controlled conditions. However, it is necessary to extend these types of studies and increase the monitoring time during the exposure in the gradient, since animals with benthic habits and little movement require greater observation times. Therefore, it is important to know the aspects of the thermoregulatory behavior in sea cucumbers to apply this knowledge in farming practices.

### 4.3. Thermal Resistance of Isostichopus isabellae

In our study, the resistance responses to temperature were evaluated, although only tests with a thermal increase from 26 °C and a decrease from 23 °C were carried out. Evidence on the influence of the acclimation temperature in these responses, even when the thermal difference between both is 3 °C, has been reported in other studies. In addition, the aim was to observe the resistance of the sea cucumber *Isostichopus isabellae* to define the responses that characterize the maximum (CTMax) and minimum (CTMin) critical temperatures based on the definition proposed by Cox [72], where “The thermal limit lower or higher is, the arithmetic mean of the thermal spots in which the activity locomotive is disorganized, and the animal loses the ability to escape the condition that quickly leads to death”. In regard to this definition, the superior thermal limit stood at 36 ± 0.3 °C and was characterized by the relaxation final response with the body and tentacles extended. The lower thermal limit was at 8 ± 0.5 °C, corresponding to animals showing decreased movement, relaxed tentacles, and the cessation of body movement (DMP + DRE + CBM). Behaviors that characterize critical temperatures are used as indicators of stress and adaptation in aquatic animals, and this is the first study reporting the responses to increases and decreases in temperature to characterize the CTMax and CTMin in sea cucumbers.

The start of the activity of the cucumbers for both resistance trials was taken when there was a variation of 2.5 °C; however, the behavior in the cucumbers before the temperature variation was different during the increases and decreases in temperature characterizing the limits for CTMax and CTMin. The cucumbers exposed to an increased temperature presented increase in length and movement of the podia. Sea cucumbers exposed to a low temperature showed changes in their behavior, expressed, for example, as slow movement. Similar behavior was reported by [24] and [53] for the sea cucumber *Apostichopus japonicus*. Also, the authors stated that a variation in water temperature caused the sea cucumbers to significantly decrease their food intake and, consequently, their growth due to variation in their metabolic rate. In our study, the animals exposed to an increase or decrease in water temperature were not fed during the trial, and taking into account that the temperature factor affects different physiological responses in aquatic organisms, it is likely that the cucumbers in this study experienced changes in their metabolism.

Meng and co-workers [23] evaluated the thermal resistance of sea cucumbers (*A. japonicus*) acclimated at two temperatures (12 and 22 °C) for 30 days. The upper lethal temperatures for the sea cucumbers acclimated to 12 °C and 22 °C were 31.8 °C (confidence interval 31.5–32.1 °C) and 30.9 °C (I.C. 30.6–31.3 °C), respectively. The results of this study emphasize that acclimation affects the upper thermal limit of sea cucumbers, and organisms that spend winter in southern China may have a greater thermal resistance than those bred in the north. The results of our study with native sea cucumbers from the Caribbean Sea exposed to increases and decreases in temperature showed that the animals acclimated at 23 °C endured temperatures below 8 ± 0.5 °C, i.e., a thermal decrease of up to 14 °C, while those acclimated at 26 °C endured temperatures of up to 36 ± 0.3 °C, with an increase range of 10 °C. The differences between these species may have been due to the environmental conditions to which they were exposed, since Japanese species, depending on the season of the year, could be exposed to temperatures that range from 0 to 30 °C. Instead, in Colombia, the sea temperatures throughout the year fluctuate between 25.6 °C and 31.1 °C. However, the different studies carried out to determine the thermal tolerance and resistance of aquatic organisms emphasize the effect of acclimation temperature on these responses.

On the other hand, there have been no studies related to the critical thermal maximum and minimum upper and lower thermal limits in sea cucumbers. Relatively similar work is attributed to Ji and co-workers [64], who assessed the physiological response and growth of *A. japonicus* when exposed to an increasing temperature from 16 °C to 26 °C or to thermal fluctuations. The authors showed that the species has a wide range of tolerance, which depends on the temperature of acclimation. Regarding culture, they found that cucumbers grown in the south of China have a higher tolerance to high temperatures than those grown in northern China, where the temperature is lower. In this study, physiological aspects of *Isostichopus isabellae* were not assessed. However, this study will give rise to future research in order to understand the biological performance of this species under different thermal scenarios.

## 5. Conclusions

The temperature of 23 °C favored the survival (86%) of the sea cucumber *Isostichopus isabellae* in captivity, and this turned out to be the best temperature for maintenance in this experiment. This temperature could be used as the initial point for the acclimation of sea cucumbers in experimental culture. Nevertheless, new studies with a broader range of temperatures will contribute to corroborating this claim. The temperature at which the cucumbers were acclimated had a direct impact on their thermal preference and thermal resistance. However, to elucidate if there is internal damage that compromises their survival in the period after returning to the collection temperature, complementary studies on the biological performance of these organisms are required.

## Figures and Tables

**Figure 1 animals-14-03613-f001:**
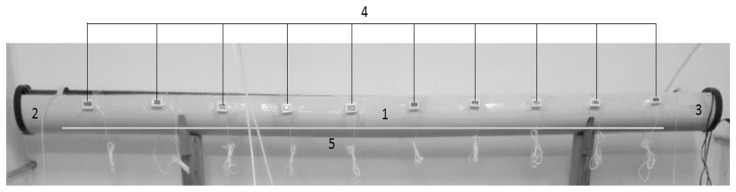
Diagram of the thermal channel for the study of the preferred temperature of sea cucumber *Isostichopus isabellae*. (1) Three-meter PVC pipe (6” Ø), (2) frozen hydrogel bags, (3) heaters (100 W), (4) digital thermometers + thermocouple, and (5) aeration line (located along the pipe).

**Figure 2 animals-14-03613-f002:**
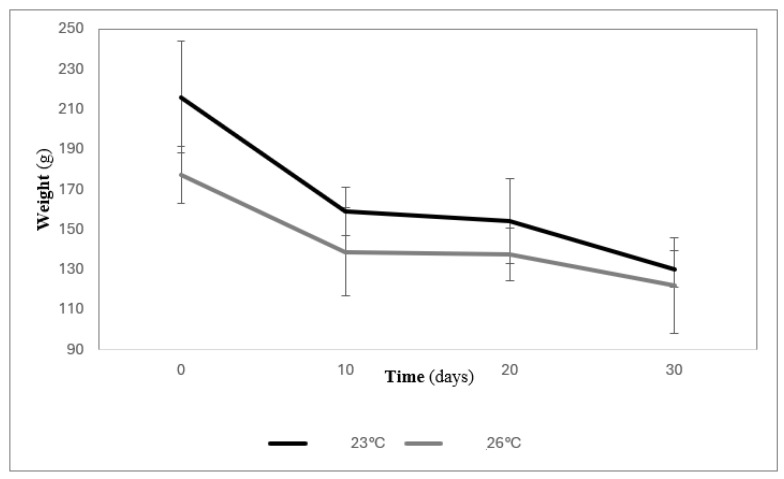
Average weight (±standard deviation) of sea cucumber *Isostichopus isabellae* acclimated to 23° and 26 °C for 30 days.

**Figure 3 animals-14-03613-f003:**
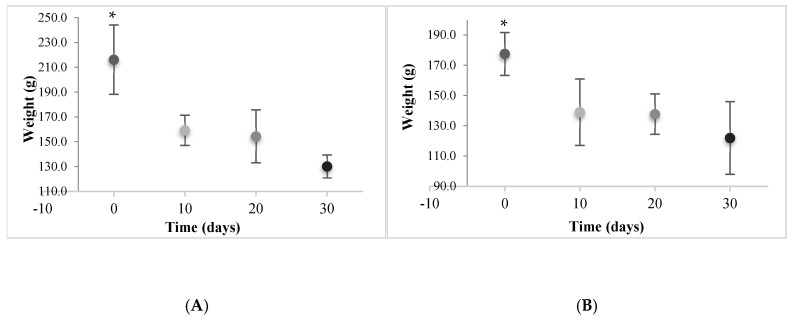
Weight loss (g) rate of sea cucumber *Isostichopus isabellae* acclimated to 23° (**A**) and 26 °C (**B**) for 30 days. (*) indicates statistically significant differences (*p* < 0.05).

**Figure 4 animals-14-03613-f004:**
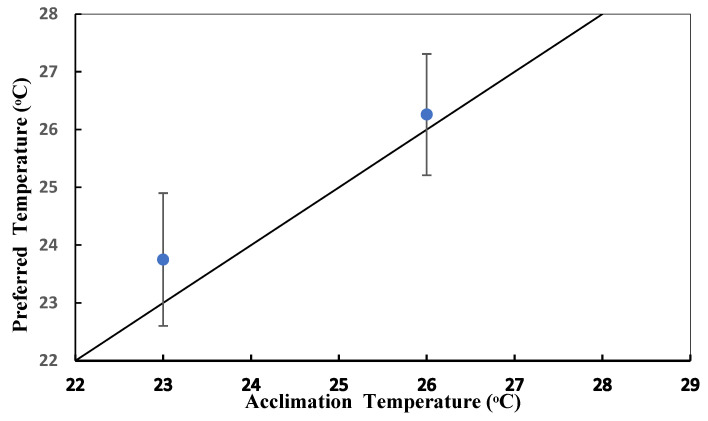
Mean values ± standard deviation of the preferred temperature of sea cucumbers *Isostichopus isabellae* acclimated at 23° and 26 °C (Student’s *t*-test; *p* < 0.001). n = 12 by temperature.

**Figure 5 animals-14-03613-f005:**
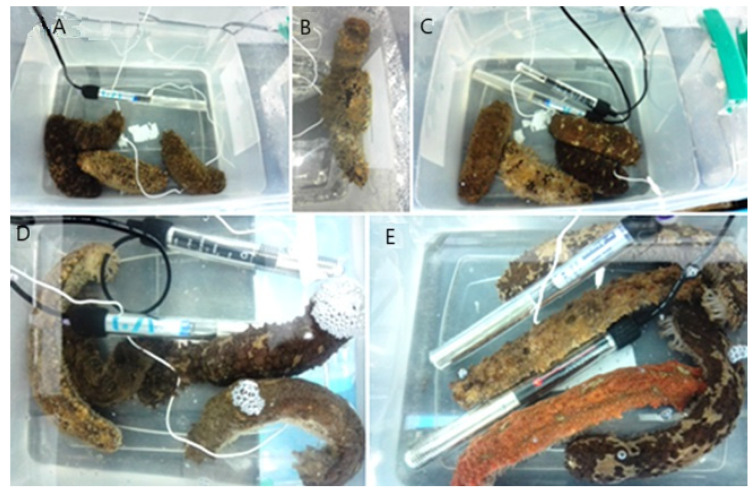
Behavior of the sea cucumber *Isostichopus isabellae* acclimated at 26 °C and exposed to an increase in the temperature. (**A**) Time 0; (**B**) Normal behavior; (**C**) Adults located on the walls of the aquarium; (**D**) Sea cucs increase in movement of podia; (**E**) Adults were exposed to seven hours of monitoring. n = 12 individuals.

**Figure 6 animals-14-03613-f006:**
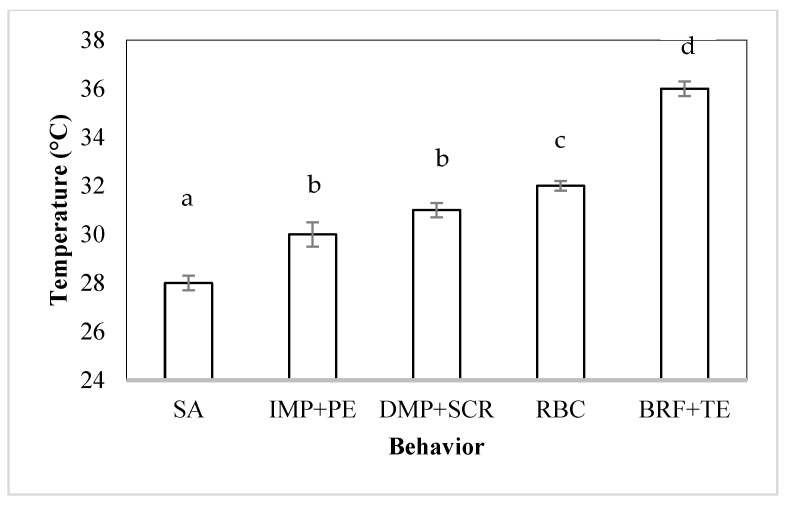
Sequence of behaviors of the sea cucumber *Isostichopus isabellae* acclimated to 26 °C and exposed to an increase in the water temperature. Adults were exposed to seven hours of monitoring. n = 12 animals. SA: start of activity, IMP + PE: increase in movement of podia and papillae extended, DMP + SCR: decrease in the movement of podia and start of corporal relaxation, RBC: relaxation and body contraction, BRF + TE: body relaxation end and tentacles extended. Letters indicate significant differences (*p* < 0.05).

**Figure 7 animals-14-03613-f007:**
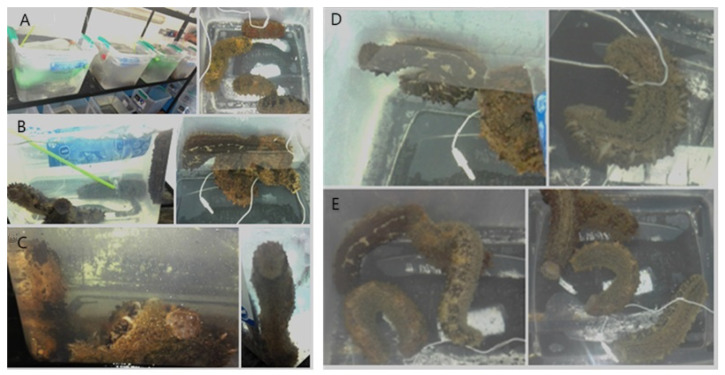
Behavior of the sea cucumber *Isostichopus isabellae* acclimated to 23 °C and exposed to a decrease in the temperature. (**A**) Normal behavior (time 0); (**B**) Sea cucs began to move toward the walls or the corner of the aquarium; (**C**) Adults overlapped with each other; (**D**) Total decrease in movement and body position in U; (**E**) Sea cucs do not move. Adults were monitored for eight hours. n = 12 animals.

**Figure 8 animals-14-03613-f008:**
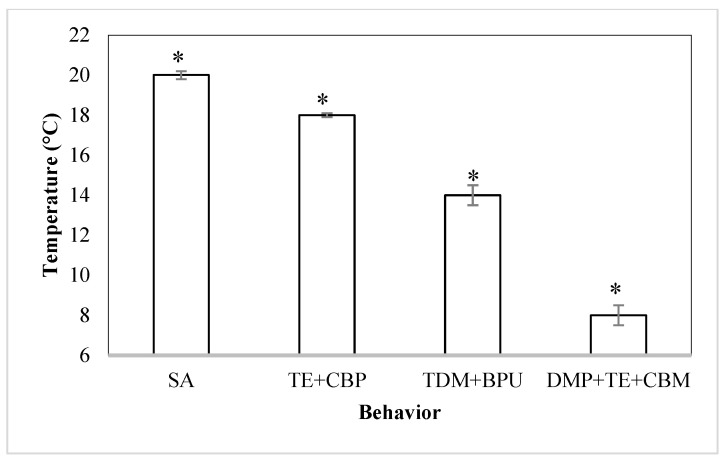
Sequences of behaviors of the sea cucumber *Isostichopus isabellae* acclimated to 23 °C and exposed to a decrease in the water temperature. Adults were monitored for eight hours. n = 12 individuals. SA: start of activity, TE + CBP: tentacles extended and body cobra position, TDM + BPU: a total decrease in the movement and U-shaped body position, DMP + TE + CBM: decrease in the podia movement, tentacles relaxed, and cessation of body movement. (*) indicates statistically significant differences (*p* < 0.05).

**Table 1 animals-14-03613-t001:** Physicochemical parameters of water during the acclimation of sea cucumbers at 23 °C and 26 °C. Median ± SD.

Water Parameters	Temperature 23 °C	Temperature 26 °C
Water temperature (°C)	23.02 ± 0.31	26.09 ± 0.12
pH	8.10 ± 0.12	8.10 ± 0.13
Dissolved oxygen (mg L^−1^)	7.21 ± 0.47	6.82 ± 0.61
Salinity (UPS)	37.13 ± 1.05	37.45 ± 1.02

**Table 2 animals-14-03613-t002:** Growth performance and survival of sea cucumber *Isostichopus* sp. aff *badionotus* maintained at the experimental temperatures of 23 °C and 26 °C. Adults exposed to 30 days of culture. *n* = 24 individuals per treatment. SGR, specific growth rate; C.V., coefficient of variation. Average ± SD.

Variables	23 °C	26 °C
Initial weight (g)	216.1 ± 29.9	177.5 ± 14.2
Final weight (g)	130.2 ± 9.1	122.0 ± 24.0
Weight gain (%)	−39.18 ± 7.9	−27.89 ± 1.5
SGR (% d^−1^)	−1.67 ± 0.4	−1.13 ± 0.7
C.V. (%)	7.0	19.7
Survival (%)	87.7	10.0

## Data Availability

The original contributions presented in this study are included in the article. Further inquiries can be directed to the corresponding author.

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
