# Peer review of "Thermal Behavior of Tropical Sea Cucumber of Isostichopus isabellae: Preliminary Issues"

_animals, 2024, doi:10.3390/ani14243613_

Round 1

Reviewer 1 Report

Comments and Suggestions for Authors

The study is well-defined, investigating the growth, thermal preference and thermal tolerance of Isostichopus isabellae. This is an important contribution to the understanding of the biology of a tropical species with ecological and commercial value.

The experimental design is broad and includes multiple aspects, such as thermal tolerance tests (CTMax and CTMin), thermal preference assessments and the use of various physicochemical parameters, ensuring a thorough investigation.

The results provide valuable information on the optimal maintenance temperatures of Isostichopus isabellae, useful for the development of captive breeding programs of this species.

The authors cited several relevant studies to support their findings, especially in the field of thermal biology and sea cucumber aquaculture, making the paper well-founded in the scientific literature.

The application of various statistical tests (t-test, ANOVA) to compare the effects of different temperatures on sea cucumbers adds rigor to the study and allows reliable conclusions to be drawn.

The study only investigated two temperatures (23°C and 26°C), which may not be sufficient to draw comprehensive conclusions about the thermal biology of the species. It is recommended that the study be extended to a wider range of temperatures as this may provide more robust results.

The thermal preference tests were conducted for only 4 hours, which may not be sufficient to observe the full range of behavioral responses, especially for slow-moving species such as sea cucumbers. Extending the observation period may provide more accurate insights.

The article does not mention long-term effects on the survival or health of sea cucumbers after returning to natural habitat temperatures. Including such data would strengthen the conclusions about thermal tolerance and long-term survival.

The authors mention that sea cucumbers lost weight, probably due to the diet provided, but this aspect was not explored further. A more detailed analysis of the species' dietary requirements and their impact on growth at different temperatures would improve the article.

The paper provides valuable insights into the thermal behavior of Isostichopus isabellae and is highly relevant to aquaculture. However, expanding the scope of the research, especially with a wider temperature range, longer observation times, and more detailed analysis of dietary requirements, could further strengthen the conclusions. Grammatical issues are minimal, but should be addressed for clarity and readability.

The following grammatical corrections are proposed

In the Abstract:

Original: "We do not find differences in average weight loss between the two experimental temperatures."

Correction: "We do not find differences in average weight loss between the two experimental temperatures."

Section Introduction:

Original: "Water temperature is one of the most important factors that affect growth..."

Edit: "Water temperature is one of the most important factors that affect growth..."

Methods:

Original: "Sea cucumbers were acclimated to two temperatures..."

Edit: "Sea cucumbers were acclimated to two temperatures..."

Results:

Original: "The weight loss in the first 10 days showed statistically significant differences..."

Edit: "The weight loss in the first 10 days showed statistically significant differences..."

Discussion:

Original: "These results agree with those reported by..."

Edit: "These results agree with those reported by..."

Comments on the Quality of English Language

The following grammatical corrections are proposed

In the Abstract:

Original: "We do not find differences in average weight loss between the two experimental temperatures."

Correction: "We do not find differences in average weight loss between the two experimental temperatures."

Section Introduction:

Original: "Water temperature is one of the most important factors that affect growth..."

Edit: "Water temperature is one of the most important factors that affect growth..."

Methods:

Original: "Sea cucumbers were acclimated to two temperatures..."

Edit: "Sea cucumbers were acclimated to two temperatures..."

Results:

Original: "The weight loss in the first 10 days showed statistically significant differences..."

Edit: "The weight loss in the first 10 days showed statistically significant differences..."

Discussion:

Original: "These results agree with those reported by..."

Edit: "These results agree with those reported by..."

Author Response

Thank you very much for taking the time to review this manuscript and express our gratitude for the feedback. Please find the detailed responses below and the corresponding revisions/corrections in track changes in the re-submitted files.

Reviewer 1

Comments 1: The study only investigated two temperatures (23°C and 26°C), which may not be sufficient to draw comprehensive conclusions about the thermal biology of the species. It is recommended that the study be extended to a wider range of temperatures as this may provide more robust results.

Response  1:

We appreciate the evaluator's suggestion, which is highly valued. A preliminary study has been conducted at two temperatures, and further investigations will be undertaken to assess additional temperatures in order to identify the optimal maintenance temperature for the cultivation of tropical wild sea cucumber species. As we refine the production cycle of sea cucumbers under controlled conditions, there remains much to explore.

Comments 2: The thermal preference tests were conducted for only 4 hours, which may not be sufficient to observe the full range of behavioral responses, especially for slow-moving species such as sea cucumbers. Extending the observation period may provide more accurate insights. 

Response  2:

The evaluator's intent is acknowledged. Nevertheless, the animals underwent a 30-day acclimatization period, which suggests that these preliminary data may provide an approximate understanding. However, further experiments with an extended evaluation period will yield definitive information about sea cucumber thermal biology.

Comments 3: The article does not mention long-term effects on the survival or health of sea cucumbers after returning to natural habitat temperatures. Including such data would strengthen the conclusions about thermal tolerance and long-term survival. 

Response  3:

Regarding thermal tolerance, the experiment was stopped when 50% of the sea cucumbers ceased their podia movement. This sign indicates the death of the organisms due to the loss of mobility in their podia. Although different temperatures were not examined, a substantial amount of preliminary information was gathered, which is lacking in tropical species of sea cucumber. Therefore, this provides a solid base for further studies of this nature.

Comment 4: The authors mention that sea cucumbers lost weight, probably due to the diet provided, but this aspect was not explored further. A more detailed analysis of the species' dietary requirements and their impact on growth at different temperatures would improve the article.

Response  4:

The authors have adhered to the dietary requirements outlined by other researchers studying sea cucumber species, which include microalgae, sand of varying thicknesses, sediment, and pelletized algal powder. In this regard, efforts have been made to replicate the conditions in a laboratory setting. However, given that this is a relatively new area of research, it is essential to explore various types of diets (both macro and microalgae), sediments, and artificial food sources to achieve an optimal nutritional level.

Comment 5: The paper provides valuable insights into the thermal behavior of Isostichopus isabellae and is highly relevant to aquaculture. However, expanding the scope of the research, especially with a wider temperature range, longer observation times, and more detailed analysis of dietary requirements, could further strengthen the conclusions. Grammatical issues are minimal, but should be addressed for clarity and readability.

Response  5:

The authors express thankfulness for the reviewer’s comments. In fact, a new experiment involving juveniles obtained in the laboratory under varying thermal conditions is nearing completion, which will enhance the understanding of the biological information and thermal requirements of tropical sea cucumber species.

 The grammatical corrections were made as requested by the evaluator.

The following grammatical corrections are proposed

In the Abstract:

Original: "We do not find differences in average weight loss between the two experimental temperatures."

Correction: "We do not find differences in average weight loss between the two experimental temperatures."

Section Introduction:

Original: "Water temperature is one of the most important factors that affect growth..."

Edit: "Water temperature is one of the most important factors that affect growth..."

Methods:

Original: "Sea cucumbers were acclimated to two temperatures..."

Edit: "Sea cucumbers were acclimated to two temperatures..."

Results:

Original: "The weight loss in the first 10 days showed statistically significant differences..."

Edit: "The weight loss in the first 10 days showed statistically significant differences..."

Discussion:

Original: "These results agree with those reported by..."

Edit: "These results agree with those reported by..."

Reviewer 2 Report

Comments and Suggestions for Authors

The authors present an interesting study on the growth and thermal behavior of Isostichopus isabellae. However, I have a major concern regarding the framing of the paper.

In all tested conditions, the sea cucumbers experienced substantial weight loss. This suggests that none of the tested temperature regimes provide an optimal maintenance condition for this species. It may be more appropriate for the authors to frame the study as an exploration of the physiological limits or stress responses of Isostichopus isabellae, rather than identifying optimal conditions for maintenance or growth. Given the weight loss, it is clear that these conditions do not represent opportunistic environments for this species. A discussion is needed to address why the weight loss occurred—was it due to stress, food scarcity, or some other factor?

Here are some other minor comments:

1. The authors present valuable data on thermal preference and resistance. However, it would be helpful to link this information to practical applications, particularly in aquaculture or conservation. How do these thermal preferences align with natural environmental conditions in the wild? Does the sea cucumber have a favorable growth season in the wild?

2. The paper briefly mentions the provided diet of sediment and Tetraselmis sp. It would be useful to discuss whether this diet meets the nutritional requirements of the species, especially since the weight loss was substantial. Could alternative food sources be considered?

Author Response

Comments 1: The authors present an interesting study on the growth and thermal behavior of Isostichopus isabellae. However, I have a major concern regarding the framing of the paper.

In all tested conditions, the sea cucumbers experienced substantial weight loss. This suggests that none of the tested temperature regimes provide an optimal maintenance condition for this species. It may be more appropriate for the authors to frame the study as an exploration of the physiological limits or stress responses of Isostichopus isabellae, rather than identifying optimal conditions for maintenance or growth. Given the weight loss, it is clear that these conditions do not represent opportunistic environments for this species. A discussion is needed to address why the weight loss occurred—was it due to stress, food scarcity, or some other factor?

Response 1:

Thank you very much for taking the time to review this manuscript. Please find the detailed responses below:

The authors have adhered to the dietary requirements outlined by other researchers studying sea cucumber species, which include microalgae, sand of varying thicknesses, sediment, and pelletized algal powder. However, given that this is a relatively new area of research, it is essential to explore various types of diets (both macro and microalgae), sediments, and artificial food sources to achieve an optimal nutritional level.

The authors acknowledge the reviewer’s comments and propose a modification to the article's title, placing greater emphasis on thermal behavior rather than growth rates. They also consider that the experimental sea cucumbers were adults, which means their growth rate would not be significantly represented in a graph; however, their response to thermal stress would be evident. Consequently, the revised title is: Thermal behavior of the tropical sea cucumber of Isostichopus isabellae: Preliminary issues.

Here are some other minor comments:

Comments 2: The authors present valuable data on thermal preference and resistance. However, it would be helpful to link this information to practical applications, particularly in aquaculture or conservation. How do these thermal preferences align with natural environmental conditions in the wild? Does the sea cucumber have a favorable growth season in the wild?

Response 2:

Sea cucumber breeding season takes place in the second half of the year (from July to November (Agudelo & Rodríguez, 2015), when the seawater is warmer due to the seasonal rains.

In their natural habitat, the tropical cucumbers assessed in this study thrive at temperatures ranging from 20 to 26°C, typically at depths between 5 and 10 meters. When acclimatized under controlled conditions for aquaculture practices, they appear to experience most favorable well-being at approximately 23°C. This temperature may represent an ideal starting point for larger-scale cultivation efforts. A broader temperature range would provide a more practical conclusion, enhancing the overall development of the species.

Comments 3: The paper briefly mentions the provided diet of sediment and Tetraselmis sp. It would be useful to discuss whether this diet meets the nutritional requirements of the species, especially since the weight loss was substantial. Could alternative food sources be considered?

Response 3:

In nature, animals feed on detritus, sand, and microalgae, a diet that is replicated in a controlled environment. New studies are being conducted to explore diets that may alleviate the nutritional issues identified during the cultivation of sea cucumbers, which are beginning to be implemented. This research employed diets referenced by other authors for sea cucumbers.

One potential enhancement could involve the provision of specially formulated artificial diets for marine animals, combined with the aforementioned nutrients.

Reviewer 3 Report

Comments and Suggestions for Authors

Manuscript ID: animals-3211861 needs changes in terms of:

1. Structures:

- graphic form (writing with the full name of the species, organization of figure 3 on the page in compliance with the presentation requirements for a scientific work, abbreviations to be revised, lines 58-61);

- the authors are encouraged to revise the structure of the Materials and Methods chapter, thus clearly and distinctly presenting the materials used and at the same time separately the methods, the methodologies used, eliminating the content elements that are the subject of other chapters (Results, Discussions). 

2. The scientific content of the paper:

- We also encourage the authors that the description of the experiment/experiments also present the control sample/s for each research scenario as well as the clarification of the research objectives for each proposed research scenario.

- the Materials and Methods chapter must be introduced in the work with a distinct presentation of the materials used and also of the working methods that were the basis of the conception of the presented manuscript.

- Subchapter 2.1 Ethics statement, we believe that it must be developed in such a way as to respect the customs of a scientific work.

- Authors are encouraged depending on the changes to make the correlations of the newly introduced scientific arguments by presenting them in the Results, Discussions and Conclusions chapters;

- Authors must also insert the hyperlinks to the bibliography and, if necessary, correlate the bibliography after restructuring the manuscript.

Author Response

We appreciate the evaluator's suggestion, which is highly valued. Thank you very much for taking the time to review this manuscript and express our gratitude for the feedback. Please find the detailed responses below and the corresponding revisions/corrections in track changes in the re-submitted files, and we look forward to your response.

Comments 1: 1. Structures:

- graphic form (writing with the full name of the species, organization of figure 3 on the page in compliance with the presentation requirements for a scientific work, abbreviations to be revised, lines 58-61);

- the authors are encouraged to revise the structure of the Materials and Methods chapter, thus clearly and distinctly presenting the materials used and at the same time separately the methods, the methodologies used, eliminating the content elements that are the subject of other chapters (Results, Discussions). 

Response 1: The authors believe that a rigorous scientific structure is being followed, consistent with the subject matter discussed. We would appreciate greater clarity regarding the specific aspects that require correction. However, our efforts have been focused on the scientific domain of the topic at hand, adhering to the established scientific standards typical of such studies.

Comments 2: The scientific content of the paper:

- We also encourage the authors that the description of the experiment/experiments also present the control sample/s for each research scenario as well as the clarification of the research objectives for each proposed research scenario.

- the Materials and Methods chapter must be introduced in the work with a distinct presentation of the materials used and also of the working methods that were the basis of the conception of the presented manuscript.

Response 2:

The authors believe that a rigorous scientific structure is being followed, consistent with the subject matter discussed.

Comments 3: - Subchapter 2.1 Ethics statement, we believe that it must be developed in such a way as to respect the customs of a scientific work.

- Authors are encouraged depending on the changes to make the correlations of the newly introduced scientific arguments by presenting them in the Results, Discussions and Conclusions chapters;

- Authors must also insert the hyperlinks to the bibliography and, if necessary, correlate the bibliography after restructuring the manuscript.

Response 3: The requests were accepted by the authors. We hope they will meet the evaluator's requirements.

Reviewer 4 Report

Comments and Suggestions for Authors

Review Report

The authors of this manuscript submitted an article titled “Growth and thermal behavior of the tropical sea cucumber of Isostichopus isabellae: Preliminary issues." The article presents a study that investigated the growth, preferences, and thermal resistance of the sea cucumber Isostichopus isabellae to understand its thermal biology. Sixty Isostichopus isabellae were kept in tanks at two temperatures (23°C and 26°C) for 30 days to determine the optimal maintenance temperature.

The findings in the manuscript were credible, but it cannot be published in animals in this form until after a minor revision centred on the following points:

Major Points

  1. The authors of this manuscript should thoroughly review the sentences that refer to the supporting references made to corroborate their findings. Sorry, this is a bad writing style, and it was used all through the manuscript. The name of the authors in these statements should be mentioned, not represented by in-text citation. For example, …..(25), evaluated the effects of different rearing temperatures on growth, metabolic performance, and thermal tolerance of juveniles of sea cucumber….. in line numbers 79-81. It can be reworked as... “Zang and co-workers evaluated the effect of different rearing temperatures.”. However, it may be necessary to only paraphrase the literature texts with appropriate referencing without mentioning the authors' names.
  2. The reference style of the animals was not adhered to in the preparation of this manuscript. The author should review all the references and ensure instructions for authors are followed in line with animals reference format. This is also applicable to the in-text citation references; they should be in square brackets.
  3. The conclusion section should be reviewed: Words such as “new trials” are not appropriate here. I hope the authors meant... “The temperature of 23 °C favoured the survival (86%) of sea cucumber Isostichopus isabellae in captivity, and this turned out to be the optimum temperature of maintenance.”

Minor points

  1. What do you mean by this statement? “This depends on the individual's previous experience (4;5).”
  2. Please cite the correct original article of the authors of this statement in line numbers 48-52... ”This temperature has been defined by Fry (1947, p. 24) as the final preferendum: a temperature around which all individuals [of a given species] will ultimately congregate, regardless of their thermal experience before being placed in the gradient," and "that temperature at which the preferred temperature is equal to the acclimation temperature."
  3. The guidelines for the care and use of laboratory animals, such as sea cucumbers, as stated by the authors in line number 111 should align with the internationally acceptable standards, and references should be provided.
  4. Colombia is a party to the NAGOYA protocol on access and benefit sharing. Could you please include information on sample depositions and voucher numbers for the samples?
  5. Who curates the sea cucumbers? Information on the sample’s identification should be explained in sub-section 2.2 Biological samples.
Comments on the Quality of English Language

The English language requires moderate improvement.

Author Response

Thank you very much for taking the time to review this manuscript and express our gratitude for the feedback. Please find the detailed responses below and the corresponding revisions/corrections in track changes in the re-submitted files, and we look forward to your response.

Comments 1: The authors of this manuscript should thoroughly review the sentences that refer to the supporting references made to corroborate their findings. Sorry, this is a bad writing style, and it was used all through the manuscript. The name of the authors in these statements should be mentioned, not represented by in-text citation. For example, …..(25), evaluated the effects of different rearing temperatures on growth, metabolic performance, and thermal tolerance of juveniles of sea cucumber….. in line numbers 79-81. It can be reworked as... “Zang and co-workers evaluated the effect of different rearing temperatures.”. However, it may be necessary to only paraphrase the literature texts with appropriate referencing without mentioning the authors' names.

Response 1:

We apologize for the errors and inaccuracies. We have made all necessary adjustments and look forward to addressing all the requirements of the esteemed evaluator.

Comments 2: The reference style of the animals was not adhered to in the preparation of this manuscript. The author should review all the references and ensure instructions for authors are followed in line with Animals reference format. This is also applicable to the in-text citation references; they should be in square brackets.

Response 2:

We apologize for the errors and inaccuracies. We have made all necessary adjustments and look forward to addressing all the requirements of the esteemed evaluator.

Comments 3: The conclusion section should be reviewed: Words such as “new trials” are not appropriate here. I hope the authors meant... “The temperature of 23 °C favoured the survival (86%) of sea cucumber Isostichopus isabellae in captivity, and this turned out to be the optimum temperature of maintenance.”

Response 3:

We acknowledge and respect the reviewer’s comments and suggestions, which we fully accept:

“The temperature of 23 °C favored the survival (86%) of sea cucumber Isostichopus isabellae in captivity, and this turned out to be the best temperature of maintenance in this experiment. This temperature could be used as the initial point to the acclimation of sea cucumbers in experimental culture”.

Comments 4: Minor points

What do you mean by this statement? “This depends on the individual's previous experience (4;5).”

Response 4:

This concerns to the animal's prior thermal history, specifically before it is subjected to the experiment. It is essential to consider that animals may originate from either their natural habitat or captivity, and this history is crucial when conducting experimentation.

Comments 5: Please cite the correct original article of the authors of this statement in line numbers 48-52... ”This temperature has been defined by Fry (1947, p. 24) as the final preferendum: a temperature around which all individuals [of a given species] will ultimately congregate, regardless of their thermal experience before being placed in the gradient," and "that temperature at which the preferred temperature is equal to the acclimation temperature."

Response 5:

We acknowledge and respect the reviewer’s comments and suggestions, which we fully accept.

Comments 6: The guidelines for the care and use of laboratory animals, such as sea cucumbers, as stated by the authors in line number 111 should align with the internationally acceptable standards, and references should be provided.

Response 6:

The requests were accepted by the authors. We hope they will meet the evaluator's requirements.

Comments 7: Colombia is a party to the NAGOYA protocol on access and benefit sharing. Could you please include information on sample depositions and voucher numbers for the samples?

Response 7:

We have not previously considered this aspect in our studies. We hold a deep respect for nature, natural resources, and the communities surrounding them. There are national permits for the research of the species that align with the principles of the Nagoya Protocol; however, we do not utilize them directly and we cannot include information on sample depositions and voucher numbers for the samples. Should there be any restrictions on the publication of this article as a result, we will understand.

Comments 8: Who curates the sea cucumbers? Information on the sample’s identification should be explained in sub-section 2.2 Biological samples.

Response 8:

We have collaborated with Dr. Igor Eeckhaut (University of Mons, Belgium) for over ten years, during which he has served as our advisor and consultant, and he is recognized as a global expert in the field of sea cucumbers.

As hand out in the text: “The specimens came from species that were recorded previously in the Center of Biological Collections of the Universidad del Magdalena (Santa Marta, Magdalena, Colombia) (CBUMAG:ECH:00001, CBUMAG:ECH:00002,  CBUMAG:ECH:00003). Its morphology was examined and compared, and their genetic structures was acknowledged through 16S rDNA and COI data [39].”

Round 2

Reviewer 2 Report

Comments and Suggestions for Authors

I don't think the authors has addressed my major concern:

In all tested conditions, the sea cucumbers experienced substantial weight loss. This suggests that none of the tested temperature regimes provide an optimal maintenance condition for this species. Given the weight loss, it is clear that these conditions do not represent opportunistic environments for this species. A discussion is needed to address why the weight loss occurred—was it due to stress, food scarcity, or some other factor?

Author Response

Dear reviewer,

In this study, the experimental temperatures correspond to thermal conditions that cucumbers experience at some time of the year. We believe that this is the temperature at which these organisms can be maintained in the laboratory for future studies.

The results of the weight loss of the animals may be due to the type of feeding, since some studies have highlighted that several environmental factors such as salinity, temperature, and biological factors such as population density and type of feeding, influence the growth and survival of juveniles of other species of sea cucumber. However, studies with these approaches in adult organisms are scarce, therefore, future research aims to continue generating knowledge on the various topics that are most frequently studied in juveniles.

We thank the reviewer for their important comments that have contributed to improving the work.

Kind regards